# Development of Pomegranate Peel Extract and Nano ZnO Co-Reinforced Polylactic Acid Film for Active Food Packaging

**DOI:** 10.3390/membranes12111108

**Published:** 2022-11-06

**Authors:** Lu Dai, Runli Li, Yanmin Liang, Yingsha Liu, Wentao Zhang, Shuo Shi

**Affiliations:** 1Department of Biological Engineering, Yangling Vocational and Technical College, Xianyang 712100, China; 2College of Chemistry & Pharmacy, Northwest A&F University, Xianyang 712100, China; 3College of Food Science and Engineering, Northwest A&F University, Xianyang 712100, China

**Keywords:** polylactic acid, ZnONPs, pomegranate peel extract, packaging

## Abstract

The multifunctional packaging used for fresh food, such as antioxidant and antimicrobial packaging, can reduce food waste. In this work, a polylactic acid (PLA)-based composite film with antioxidant and antibacterial properties was prepared by using nano-zinc oxide (ZnONPs) and pomegranate peel extract (PEE) via the solvent-casting method. Different amounts of PEE (0.5, 1, 1.5 and 2 wt%) and 3 wt% ZnONPs were added to PLA to produce the active films. The results of various characterizations (SEM, XRD, etc.) showed that ZnONPs and PEE were uniformly dispersed in PLA film. Compared to PLA films, the PLA/ZnONPs/PEE films showed an increased UV barrier, water vapor permeability and elongation at break, and decreased transparency and tensile strength. In addition, the antioxidant activity of the composite film was evaluated based on DPPH and ABTS. The maximum DPPH and ABTS scavenging activities of PLA/ZnONPs/PEE were 96.2 ± 0.8% and 93.1 ± 0.5%. After 24 h, PLA/ZnONPs/PEE composite film inhibited 1.4 ± 0.05 Log CFU/mL of *S. aureus* and 8.2 ± 0.35 Log CFU/mL of *E. coli*, compared with the blank group. The results showed that PLA/ZnONPs/PEE composite film had good antibacterial and antioxidant activities. Therefore, the composite film showed great potential for food packaging.

## 1. Introduction

The main role of food packaging is to protect food from outside influence and damage, accommodate food, and provide composition and nutritional information to consumers [1]. In recent years, plastics have been used more and more widely in packaging. Inert and non-biodegradable plastic materials account for 30% of municipal solid waste [2]. Several biopolymers including starch, cellulose, chitosan, polylactic acid (PLA), polycaprolactone (PCL), and polyhydroxybutyrate (PHB) are widely used in food packaging [3]. PLA is a linear aliphatic thermoplastic polyester found in sugar beets and cornstarch that gains usability [4] through bacterial fermentation. It is considered to be one of the most promising biobased and biogradable polymers for food packaging, because of its comparable mechanical properties with similar petroleum-based polymers such as polyethylene and polypropylene. In addition, PLA is safe for food packaging applications because it has proven to be “generally recognized as safe” (GRAS) [5,6]. However, pure PLA film has almost no antibacterial and antioxidant activities, which limits its application in packaging. At present, the mechanical properties and permeability of PLA can be improved by adding plasticizers, polymer blends, conjugates, copolymers, and nanotechnology. Nanometals are considered to be one of the most direct and effective strategies to improve antimicrobial and UV properties [7,8].

Zinc oxide nanoparticles (ZnONPs) are compound semiconductor materials, the characteristics of which include high stability, high ultraviolet (UV) absorption capacity, broad antibacterial spectrum, long antibacterial duration, easy decomposition at high temperature, and low cost [9,10]. ZnONPs have been considered potential candidates for polymer matrix-enhancing materials [11] in food packaging applications. The addition of ZnONPs to PLA effectively improved the barrier properties, UV barrier, and mechanical properties, and increased its antibacterial activity [9,12]. The antioxidant activity of food packaging film is also an important factor to maintain food quality and prolong the shelf life of packaged food. Excellent antioxidant efficiency can delay adverse biological or chemical changes of certain microorganisms in foods. A successful combination of both natural extracts and nanoparticles is a research direction of the field of active nanobiocomposites [13,14]. Many active packaging forms have been studied, for example, propolis and its byproducts are added into natural polymers such as pectin to prepare antibacterial and antioxidant packaging films [15,16]. Pomegranate peel extract (PEE) is one of the most valuable byproducts in the food industry, and a rich source of bioactive compounds [17]. A large number of studies have shown that PEE is rich in a variety of phenolic substances, among which ellagic acid (EA) is one of the main active components. It has significant biological activities, such as antioxidant, anti-tumor, anti-inflammatory, antiviral neuroprotection, and antibacterial properties, among other properties [18,19]. In previous studies, PEE has been incorporated into food packaging films. Seyed HadiPeighambardous et al. combined chitosan nanoparticles coated with pomegranate peel extract with gelatin to prepare active packaging films [20,21,22].

However, the use of the combination of PEE and ZnONPs to obtain PLA-based active nanocomposite films with antioxidant and antimicrobial properties has not been investigated. Therefore, in this study, PEE and ZnONPs were added to PLA as functional substances to prepare PLA-based antioxidant and antibacterial composite films. The morphological structure, physical properties, as well as antioxidant and antibacterial activity were investigated. Finally, this versatile film was applied to extend the shelf life of cherry tomatoes (Figure 1).

## 2. Materials and Methods

### 2.1. Materials

Polylactic acid (PLA) was purchased from Zhejiang Haizheng Biomaterial Co., Ltd. (Taizhou, China). Zinc oxide nanoparticles (ZnONPs) were purchased from Shanghai source leaf purchase (Shanghai, China). Pomegranate peel extract (PEE) was purchased in Rena biology co, (Xi’an, China). Luria-Bertani (LB) agar and broth were purchased from Beijing Land Bridge Technology Co., Ltd. (Beijing, China). Then, 2,2-diphenyl-1-picrylhydrazyl (DPPH), 2,2′-azinobis (3-ethylbenzothiazoline-6-sulfonic acid) (ABTS) were purchased from Sigma-Aldrich (St. Louis, MO, USA), and cherry tomatoes were purchased Hao You Duo supermarket (Yangling, China). The strains used were *Escherichia coli* (*E. coli* ATCC 25922) and *Staphylococcus aureus* (*S. aureus*, ATCC 25923).

### 2.2. Preparation of PLA-Based Composite Films

In this experiment, PLA-based composite film was prepared with chloroform as solvent. First, PLA (1.5 g) was dissolved in 50 mL chloroform, stirred with a magnetic stirrer for 2 h until PLA was completely dissolved. Then ZnONPs (3 wt% PLA) were dispersed into the PLA mixed solution by ultrasonic for 30 min. Different concentrations of PEE (0.5, 1, 1.5, and 2 wt% PLA) were added and stirred for 2 h, and conducted with ultrasonic treatment for 30 min to uniformly distribute PEE in the solution. After stirring for 2 h, 10 mL of the film-forming solution was evenly poured onto a 90mm diameter glass petri dish. It was then dried in a fume hood at room temperature (25 °C) for 24 h before the incompletely dried composite film was transferred to an oven for another 24 h of drying at 60 °C, where it was then peeled off the glass plate. The composite films were designed as PLA/ZnONPs, PLA/ZnONPs/PEE_0.5_, PLA/ZnONPs/PEE_1_, PLA/ZnONPs/PEE_1.5_ and PLA/ZnONPs/PEE_2_. The control PLA film was prepared by the same method without adding PEE and ZnONPs.

### 2.3. Characterization of PLA Based Composite Films

#### 2.3.1. Morphology of Surface and Cross-Section

The macro-morphology of the PLA based composite film was observed by using a digital camera (Canon EOSM2, Tokyo, Japan), and the film was studied in a scanning electron microscope (SEM, Hitachi S-4800, Tokyo, Japan) to understand the microstructure of its surface and cross-section. Prior to analysis, vacuum-coated gold sputtering was used to improve its conductivity. For the cross-section, the film was dried and fractured with liquid nitrogen [23].

#### 2.3.2. FTIR Spectroscopy and XRD Analysis

The chemical structure of the thin film samples was studied based on a FTIR spectrometer (Vetex70, Karlsruhe, Germany) [24]. The film sample was placed (2 × 2 cm) on the ray exposing stage. The crystallinity was analyzed by using an X-ray diffraction using Cu Kα radiation at a voltage of 40 kV and a current of 30 mA. The refracted radiation of the sample was recorded at ambient temperature in the range of (2θ) from 5° to 80° and scanned at a diffraction angle of 2θ. The scanning speed was 10°/min (Bruker D8 propulsion A25, Karlsruhe, Germany) [25].

#### 2.3.3. Thermal Stability

The thermal stability of the PLA based composite film samples was determined by using a thermal stability analyzer (Shimadjin TA-60WS + DTG-60A, Kyoto, Japan). The film sample (1 × 3 cm) was heated in a nitrogen atmosphere (30–700 °C, 10 °C/min). The maximum decomposition temperature was calculated by the derivative of the curve (DTG) of the TGA curve, and the weight loss rate (%) is determined by the TGA curve.

#### 2.3.4. Optical Property

UV-visible spectra based on the PLA composite films were collected in the wavelength range of 200–800 nm by a double beam UV–vis spectrophotometer (MAPADA P7, Shanghai, China). The light transmittance (%) at 280 nm and 660 nm were respectively recorded to evaluate the UV barrier and transparency properties.

#### 2.3.5. Mechanical Properties

The film thickness was measured with a digital micrometer (precision of 0.001 mm) (Aladdin, Shanghai, China) at five randomly selected locations on the surface and their average value was used. The tensile strength (TS) and the elongation at break (EB) of the film were measured by stretching the method-testing facility. The films (0.5 × 3 cm) were tightly fixed at both ends of the force arm. The tensile speed was set to 5.0 mm/s, and the corresponding tensile strength and distance were recorded. The EB of films was calculated as follows,
EB (%) = (L − L_0_)/L_0_,(1)
where L_0_ (mm) is the initial length of films and L (mm) is the length of the films at the time of break.

#### 2.3.6. Moisture Content and Water Solubility

For the calculation of moisture content of the films, the film samples (0.5 × 0.5 cm) were weighed for their initial weight (W_1_). To get the corresponding dry weight (W_2_), the film samples were dried at 105 °C. The moisture content of the film (MC) was calculated by using the following equation,
MC (%) = (W_1_ − W_2_)/W_1_ × 100,(2)
where W_1_ (g) and W_2_ (g) are the initial and corresponding dry weight of the film sample.

The film sample (0.5 × 0.5 cm) was taken and dried in a 105 °C oven for 24 h to obtain the initial dry weight (W_dry,initial_). The film was then placed in a glass permeable cup (25 mm × 25 mm) with 3 mL of distilled water to completely immerse the film, closed with plastic wrap, and stored at room temperature for 24 h. The film samples were then dried under the same conditions to obtain the final dry weight (W_dry,final_). The water solubility of the film (WS) was calculated by using the following equation [26],
WS (%) = (W_(dry,initial)_ − W_(dry,final))_/W_(dry,initial)_ × 100,(3)
where W_dry,initial_ is the initial dry weight of the film, and W_dry,final_ is the final dry weight of the film.

#### 2.3.7. Water Vapor Permeability (WVP) and Water Contact Angle (WCA)

The WVP of the film was measured by the cup method. First, add 2 g of anhydrous calcium chloride desiccant to the glass penetration cup (25 mm × 25 mm), tightly sealed by the film. The glass penetration cup was then stored in a desiccator (relative humidity of 50%, 25 °C) after recording the initial weight. Weight changes of the cup (to the nearest 0.001 mg) were recorded as a function of time every 2 h. The WVP of the film was calculated by using the following equation,
WVP = (W × d)/(∆t × s × ∆p),(4)
where W (g) represents the weight gain of the film, d (m) represents the average thickness of the film, ∆t (h) represents the penetration time, s (m^2^) represents the penetration area, and ∆p (Kpa) represents the partial vapor pressure difference between pure water and dry air. Each film was tested three times.

Contact angle analyzer (SZ-CAMB3, Shanghai, China) was used to analyze the hydrophilic properties and hydrophobicity of the film. The film sample (0.5 cm × 2 cm) was fixed on a horizontally movable sample table. A total of 5 µL of distilled water was dropped onto the thin film surface through a syringe, and the WCA the water droplets on the thin film surface was measured. Each film was measured three times, and the average value was calculated.

### 2.4. Antioxidant Activity

For the DPPH radical scavenging assay, the thin film (40 mg) was covered with 5 mL ethanol, and then sealed and shaken at 25 °C for 30 min. Then, the film solution was centrifuged for 10 min at 6000 rpm, and the supernatant was used to test the antioxidant capacity of the film. A total of 1 mL of supernatant was mixed with 3 mL of 0.2 mM DPPH of ethanol for 30 min in the dark, and then the absorbance of the sample was measured at 517 nm by using a UV-visible spectrometer. A total of 1 mL of ethanol was mixed with 3 mL of 0.2 mM DPPH of ethanol for 30 min in the dark, and then the absorbance of the sample was measured at 517 nm. A total of 1 mL of supernatant was mixed with 3 mL of ethanol for 30 min in the dark, and then the absorbance of the sample was measured at 517 nm and the DPPH radical clearance was calculated by using the following equation,
DPPH (%) = (1 − (A_1_ − A_2_)/A_3_) × 100,(5)
where A_1_ is the absorbance of the film and DPPH, A_2_ is the absorbance of the film and ethanol, and A_3_ is the absorbance of DPPH and ethanol.

For the ABTS radical scavenging assay, a total of 2.6 mmol/L of potassium persulfate was mixed with 7.4 mmol/L of ABTS (2,2′-azino-di-[3-ethylbenzthiazoline sulfonate] solution on 1:1 volume ratio for 12 h in the dark. Thereafter, the mixture was diluted with ethanol to an absorbance value of 0.7 ± 0.2 at 734 nm. The 100-µL sample was mixed with 1 ml of the diluted solution for 6 min in dark, and then the sample absorbance was measured at 734 nm by a UV visible spectrometer (A_1_). A total of 100 µL ethanol was subjected to the same treatment to measure the absorbance (A_0_). ABTS radical clearance was calculated by using equation,
ABTS (%) = (A_0_ − A_1_)/A_0_ × 100,(6)
where A_1_ was the absorbance of ABTS of test film, A_0_ was the absorbance of ABTS of blank film.

### 2.5. Antibacterial Activity

The total viable colony count method [27] was employed to perform antibacterial tests on the PLA/ZnONPs/PEE composite films. *S. aureus* and *E. coli* were aseptically inoculated in LB broth and then cultured overnight at 37 °C. They were diluted by using 0.9% saline to an initial bacterial concentration of 10^6^ CFU/mL. Film samples (80 mg) were sterilized with UV light, and then 5 mL of the diluted bacterial suspension broth containing 80 mg of the film sample was incubated at 37 °C, 180 rpm for 24 h. Samples were taken at regular intervals (0, 12, and 24 h after incubation), appropriately diluted, and coated on LB agar plates to identify viable colonies. To compare the antimicrobial activity, the same operation was performed by using the bacterial suspension without the film sample as control. The antimicrobial activity was performed in triplicate.

### 2.6. Evaluation of Cherry Tomatoes’ Preservation

When the concentration is 0.5 wt%, the effect of PEE on cherry tomatoes can be minimized, and it has enough antibacterial and antioxidant activities. Thus, the pure PLA film, the PLA/ZnONPs film, and the PLA/ZnONPs/PEE_0.5_ film were finally selected for the packaging test to assess whether the composite film can prolong the storage time of cherry tomatoes. Then cherry tomatoes of similar quality and size were selected to be sterilized in 0.01% sodium hypochlorite solution, washed with deionized water, and dried at room temperature. The cherry tomatoes were wrapped with the prepared PLA film, PLA/ZnONPs and PLA/ZnONPs/PEE_0.5_ composite film. Unpackaged cherry tomatoes were marked as blank. All cherry tomatoes are kept at room temperature. With the extension of storage time, the appearance changes of cherry tomatoes were recorded with a digital camera every two days. The weight loss was measured by weighing the difference of samples before and after the storage period, and the percentage of weight loss was calculated by the following formula,
Weight loss = (W_i_ − W_f_)/W_i_ × 100,(7)
where W_i_ is the initial sample weight (g), and W_f_ is the sample weight after storage (g).

The pH of the homogenized sample was measured by pH meter. Finally, the hardness of cherry tomatoes on the first and last day was measured by texture analyzer.

To observe the microbial condition of the cherry tomatoes after 15 days. After 15 days, the four groups of cherry tomatoes were homogenized, and the suspension was appropriately diluted. The microbial evaluation was then performed by spread plate method.

### 2.7. Statistical Analysis

All experiments were carried out in triplicate. The experimental data analysis was conducted by using SPSS software (SPSS 26.0 for windows, SPSS Inc., Chicago, IL, USA). Duncan’s new multiple-range test was used to determine the difference of means.

## 3. Results and Discussion

### 3.1. Characterization of PLA-Based Composite Films

#### 3.1.1. Morphology

SEM was used to observe the surface and cross-section morphology of PLA composite films. As shown in Figure 1, all composite films are complete without obvious defects, which indicates that PLA has good film-forming ability. The surface of the pure PLA film was relatively smooth and uniform, which is consistent with previous research results [28]. When the addition amount of ZnONPs were 3% of PLA matrix, the surface morphology and cross-section of the film were slightly changed without any agglomeration, indicating that ZnONPs had good dispersibility in the PLA composite films. When low concentration of PEE (0.5 wt%) was added, the surface morphology and cross-section changes of the composite film were thin and small, and the surface PEE particles were uniformly dispersed in the PLA/ZnONPs matrix. With the increase of PEE content (1.5 wt%), the film surface became relatively rough, and the film cross-section also showed micropores. The same phenomenon can be observed in the curcumin/PLA composite membrane studied by SwarupRoy et al. [29].

#### 3.1.2. FT-IR Analysis

FTIR analysis was performed to detect the interaction between PLA, ZnONPs, and PEE, as shown in Figure 2a. Pure PLA films, broadband at 3654 cm^−1^ were assigned to O-H tensile vibration, and peaks at 2945–3000 cm^−1^ corresponding to −C−H− tensile, the strong characteristic peak located at 1750 cm^−1^, was attributed to the C=O ester group. The peaks at 1451 cm^−1^ and 1380 cm^−1^ belong to the symmetric and asymmetric bending oscillations of the C−H. Absorption bands corresponding to the −C−O− stretch was also observed near the 1267 cm^−1^ and 1045 cm^−1^ marks [4,23]. For the PLA/ZnONPs binary composite thin films, a weak Zn−O tensile vibration was observed at 684 cm^−1^. The strength of characteristic peaks, such as C=O stretching, −C−H− bending, and −C−O− stretching were increased compared to pure PLA whose characteristic peak positions are essentially unchanged [30]. After the introduction of PEE, all of the PLA/ZnONPs/PEE showed a similar pattern when compared to the PLA/ZnONPs, with no significant change in the peak position. Thus, a large number of peaks of the PLA and PLA-based composite films appear at the same position. Furthermore, no significant new peaks were observed, indicating that PEE and ZnONPs had a physical interaction only with the PLA rather than any chemical bonds, implying the successful preparation of the PLA-based composite film.

#### 3.1.3. XRD Analysis

The crystallographic properties of the pure PLA, PLA/ZnONPs, and PLA/ZnONPs/PEE_2_ thin films were investigated by XRD analysis. As shown in Figure 2b, the pure PLA had a positive-squared crystal structure (α-form), consisting of 103 helical chains [31]. The characteristic peaks of pure PLA films were only shown around 2θ = 20°. For PLA/ZnONPs binary composite films, characteristic peaks located at 2θ = 31.6, 34.4, 36.2, 47.4, 56.7, 62.7, and 66.2°, which corresponded to (100), (002), (101), (102), (110), (103), and (200) planes, respectively, of the hexagonal crystal of ZnONPs (JCPDS card No 36-1451) [32], which indicated that ZnONPs were successfully mixed into PLA. In addition, when PEE was added to the composite film, the position and size of the characteristic peak were basically unchanged. This showed that the addition of PEE has no obvious effect on the structure of composite films. The above results showed that these PEE and ZnONPs were successfully embedded in PLA polymer matrix.

#### 3.1.4. Thermal Stability of PLA-Based Composite Films

The thermal analysis test results for the pure PLA, ZnONPs/PLA, PEE_2_/ZnONPs/PLA thin films were shown in Figure 2c. During the thermal decomposition, process, the thin film exhibited a two-step weight reduction process. The first weight loss occurred at 100–150 °C, and was mainly attributed to the evaporation of the loose-bound water [33]. The second weight loss of PLA films occurred at 260–320 °C, and the third loss for PLA/ZnONPs and PLA/ZnONPs/PEE_2_ at 350–400 °C. The second weight loss of the film was mainly due to the degradation of PLA, ZnONPs, and PEE, which was the main weight loss of the film. TGA showed that the addition of ZnONPs and PEE reduced the thermal stability of the PLA-based films. The DTG results (Figure 2d) indicated that the pure PLA, PLA/ZnONPs, and PLA/ZnONPs/PEE_2_ films shared the same final residues at 400 °C.

#### 3.1.5. Optical Property of PLA-Based Composite Films

UV blocking is an important characteristic to be considered for polymer films used as packaging materials. The UV blocking performance and transparency of the PLA-based films were evaluated by recording UV-visible transmission spectra in the range of 200–800 nm. As shown in the Figure 3b,c, PLA films had high transmittance to both UV and visible light, indicating that the PLA films had low UV-blocking performance and high transparency. The addition of ZnONPs reduced the transmittance of the film sample in the near-UV region (200–400 nm) and in the visible light region (400–760 nm), which was due to the interference of the light channel and absorption of UV of ZnONPs. The addition of PEE increased the transmittance of the film in the visible region and reduced the transmittance in the near-UV region, thereby enhancing both the transparency and UV-blocking performance of the film. This was mainly because PEE has a UV absorption capacity while reducing the interference of the light channel of ZnONPs. With the increase of PEE content, the transparency and UV barrier changes of the film were not dependent on the PEE content, probably because of the PEE aggregation caused by the increase of PEE content. When the concentration of PEE is 0.5 wt%, the visible light transmittance of the composite film is relatively increased. The results showed that the addition of ZnONPs and PEE not only increases the UV barrier property of the film, but also does not affect the visibility of the packaging products.

#### 3.1.6. Thickness and Mechanical Properties of PLA-Based Composite Films

The TS and the EB were as shown in Figure 4a. The thickness of the pure PLA film (0.0325 ± 0.00173 mm) increased significantly (*p* < 0.05) by the addition of PEE and ZnONPs, mainly due to the increase in solid content (Table 1). The TS of the pure PLA film was 88.33 ± 2.63 Mpa, and the TS of the composite film decreased (62.52 ± 4.01 Mpa) as ZnONPs were added into the PLA-based composite film. This is possibly because the aggregation of the added ZnONPs on the PLA film led to a reduction in the interaction of ZnONPs on the film surface [28]. The addition of 0.5% PEE to the composite film the EB of this film increased slightly. This is mostly because of the moderate amount of PEE‘s good compatibility with ZnONPs and PLA [34]. When the content of PEE increased to 1%, 1.5%, and 2%, TS decreased to 52.37 ± 0.74 Mpa. The reason may be that when the content of PEE is high, the stress field in PEE area would grow, crack initiation and propagation would occur more easily, and TS would decrease [19]. In contrast, the addition of ZnONPs increased the EB of PLA film and similarly, with the increase of PEE content, the EB of the composite film gradually increased to 9.19 ± 0.35%. The results were similar to those reported by Qin et al. and Yang et al. [7,35]. The film used for food preservation needs to bear external pressure and keep its integrity, so appropriately weakening the tensile strength of the film and increasing the elongation at break can prevent the packaging film from being too brittle and breaking.

#### 3.1.7. Moisture Content and Water Solubility

The MC of PLA-based composite films were also presented in Figure 4c. The addition of ZnONPs reduced the MC of the composite films, which is mainly due to the hydrophobic nature of ZnONPs [13]. Controlling the addition amount of PEE in a certain range can improve the MC of PLA-based composite films. The MC of PLA-based composite films improved when the PEE increased from 0% to 0.5%. The MC of the film decreased when the PEE rose to 1, 1.5%, and 2% because PPE was probably more hygroscopic and altered the equilibrium of the hydrophilic points, promoting water absorption. When PPE was present, the defects on the base surface increase, such as cracks and pores, which were more vulnerable to water invasion. When the PEE content exceeded above a given threshold, the PEE particles fill the gap on the base surface, reducing the water invasion and WC [19]. As presented in Figure 4c, the addition of ZnONPs increased the solubility of the film in water. It indicated a weak cohesion between PLA and ZnONPs. The addition of PEE resulted in a slight decrease in WS of the films. In short, although the water solubility of the composite film added with PEE was slightly increased, the decrease in the water content of the PLA/ZnONPs/PEE composite film could weaken the effect on cherry tomatoes during the packaging test.

#### 3.1.8. Water Contact Angle

The hydrophobic surface properties based on PLA composite films were determined by WCA studies. As shown in Figure 4b, the WCA of the film increased after adding ZnONPs. This was because ZnONPs are a hydrophobic material, and evenly distributed on the surface of the film, resulting in a reduced wettability of the film. When PEE was added to this composite film, the WCA of the film decreased and the hydrophilic properties increased as the PEE content increased, mainly attributed to the PEE being a hydrophilic material that increased the wettability of the film.

#### 3.1.9. Water Vapor Permeability

WVP is an important property of food packaging films. Films with smaller WVP can prevent water loss of food by reducing the flow of water, and achieve the purpose of prolonging the shelf life of food. The WVP values of the PLA-based composite films were shown in Figure 4d. It can be seen from the figure that the addition of ZnONPs significantly reduced the WVP of the composite film (*p* < 0.05), indicating that the addition of ZnONPs considerably improved the water barrier capacity of the PLA film. After the addition of the PEE, the WVP of this composite film increased substantially [36], mainly because the presence of PEE may have increased the hygroscopic behavior and changed the balance of the hydrophilic points, promoting water absorption. When the PEE content was increased from 0% to 2%, the WVP of the composite film was significantly increased. This might be due to the accumulation of PEE in the film, causing the decrease of the tightness of the film.

### 3.2. Antioxidant Activity of PLA Based Composite Films

DPPH scavenging activity was used to analyze the antioxidant activity of PLA-based composite film. The reducing capacity of all samples was analyzed by using the absorbance values at 517 nm wavelength. Figure 5a showed the scavenging activity of DPPH. The pure PLA film showed negligible antioxidant activity with a free radical clearance of 8.0%. The DPPH clearance of the PLA film with 3% ZnONPs was 8.7%. When 0.5 wt% of PEE was added, the DPPH clearance increased to 26.4%. With the increasing PPE concentration, the free radical scavenging activity based on the PLA film was significantly enhanced, although the combination of PEE and ZnONPs had no synergistic effect on its antioxidant activity. These results suggested that the antioxidant activity of the composite film is mainly attributed to PEE [37,38].

The ABTS radicals have been widely used to detect the free radical scavenging ability of compounds, and thus to evaluate the antioxidant activity of the thin films. Figure 5b showed the clearance of ABTS^+^ radicals from the thin film samples. The antioxidant activity of pure PLA films and PLA films containing ZnONPs alone were negligible. After the addition of PEE, the free radical scavenging activity based on PLA film was significantly enhanced with the increase in PPE concentration. The results showed that the films containing PEE have good antioxidant potential and can improve the oxidative stability of food [33].

### 3.3. Antimicrobial Activity of PLA-Based Composite Films

The antimicrobial activity of the PLA-based composite film against two strains (*S. aureus* and *E. coli*) was shown in Figure 6. As anticipated, pure PLA films had no antibacterial activity against both bacteria, although composite films containing ZnONPs showed significant antibacterial activity against both bacteria. The inhibitory effect of PLA composite film on *S. aureus* was depicted in Figure 6a. It could be seen that pure PLA film had almost no inhibitory effect. When ZnONPs were added, the growth rate of *S. aureus* was greatly decreased, indicating that PLA/ZnONPs film had significant antibacterial activity against *S. aureus*. However, with the growth of time, the antibacterial effect weakened slightly. When PEE was mixed into the composite film, the number of *S. aureus* decreased gradually after 12 h, which might be due to the gradual release of PEE later. Polyphenol compounds in PEE also exerted antibacterial activity by inhibiting the formation of biofilm. As shown in Figure 6b, the antibacterial activity of the composite film was greatly enhanced by adding ZnONPs. After PEE was added, the number of *E. coli* decreased gradually after 12 h. And when the concentration of PEE was 0.5 wt%, the composite film inhibited 1.41 Log CFU/mL of *S. aureus* and 5.59 Log CFU/mL of *E. coli* after 24 h. which indicated that the inclusion of ZnONPs and PEE enhanced the antibacterial activity of PLA composite film.

ZnONPs antibacterial activity against *E. coli* is greater than that against *S. aureus*, which is mainly attributed to the fact that Gram-positive bacteria have thick cell wall structures containing multilayers of peptidoglycan. Gram-negative bacteria, on the other hand, are composed of complex cell wall structures within a thin layer of peptidoglycan surrounded by an outer membrane. Different cell wall structures of Gram-positive bacteria and Gram-negative bacteria result in different bacterial susceptibility, which in turn leads to different antibacterial activity of ZnONPs [14,39].

### 3.4. Evaluation of Cherry Tomatoes Preservation

To assess the feasibility of PLA/ZnONPs/PEE_0.5_ packaging films, the unpackaged, PLA films and PLA/ZnONPs film were compared, as shown in Figure 7a. Unpackaged cherry tomatoes and those packaged with pure PLA films developed local mildew in the 9 days of storage, but those packaged with PLA/ZnONPs and PLA/ZnONPs/PEE_0.5_ films remained fresh. The mildew spots became more severe after 15 days. In addition, four groups of cherry tomatoes were homogenized and tested for microorganism assay. The results (Appendix A) showed that the unpackaged and PLA-packed groups grew more microorganisms than the other two groups, indicating that these two groups were more heavily contaminated with microorganisms. These results indicated that PLA/ZnONPs and PLA/ZnONPs/PEE_0.5_ composite films exhibited good antibacterial ability in packaging application.

#### Weight Loss, Hardness, and pH

As shown in Figure 7b, all cherry tomatoes showed weight loss after storage for 15 days. The loss in mass for the unpackaged samples was 15.03 ± 1.72%, significantly higher than for the group covered with the packaging film. This was mostly caused by the respiration of the fruit and the migration of water from the fruit to the surrounding environment. The PLA-based film had a good water vapor barrier ability, which could effectively reduce the water loss of the fruit. It can be observed that the mass loss for the PLA/ZnONPs film group was less than that for the pure PLA film and PLA/ZnONPs/PEE_0.5_ groups. This also corresponds to the result of WVP. All cherry tomatoes’ pH values rose to varying degrees with longer storage times, which may be related to the decomposition of amino acids. The change of cherry tomatoes covered with PLA/ZnONPs and PLA/ZnONPs/PEE_0.5_ film is the smallest, which delayed the change of the pH of cherry tomatoes (Figure 7c). After 15 days of storage, the hardness of all cherry tomatoes gradually decreased. The hardness loss of cherry tomatoes coated with PLA/ZnONPs and PLA/ZnONPs/PEE_0.5_ films decreased significantly (Figure 7d). This is similar to the result of thymol/soybean protein isolate film used for blueberry preservation [40]. These results strongly prove that PLA/ZnONPs/PEE0.5 film has a superior freshness-keeping effect [41,42].

**Figure 7 membranes-12-01108-f007:**
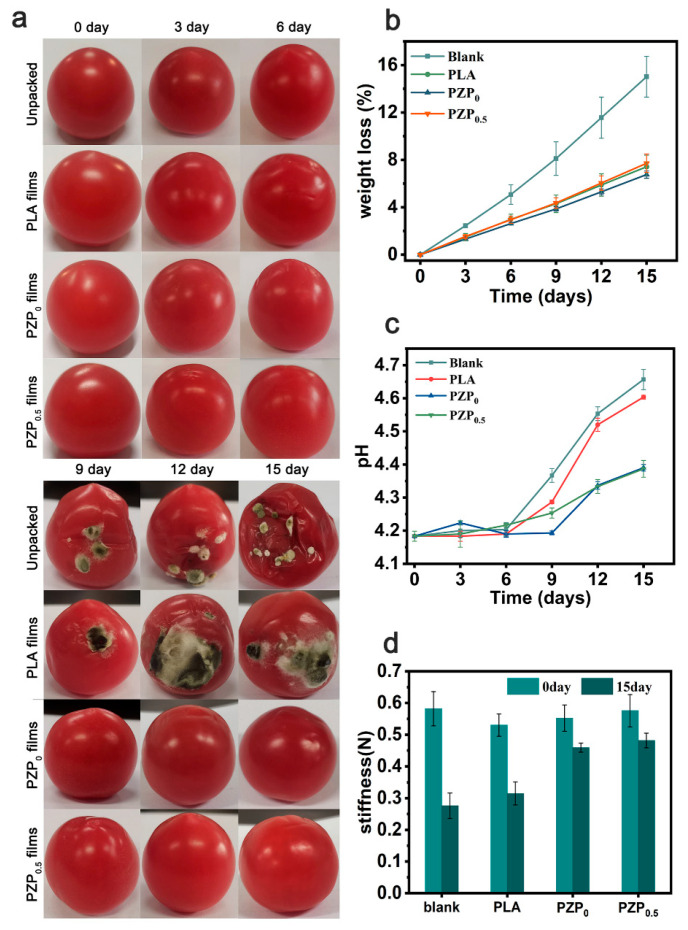
(**a**) Typical pictures of cherry tomatoes at different times with various treatments (unpacked, PLA, PZP_0_ and PZP_0.5_ film), (**b**) weight loss, (**c**) pH changes, (**d**) stiffness changes (0 and 15 days).

## 4. Conclusions

In this work, a PLA-based composite film containing ZnONPs and PEE was successfully prepared by an environmentally friendly solution casting process. The addition of ZnONPs and PEE to PLA improved the UV properties of the biopolymer film. The antibacterial test showed that the composite film with the addition of ZnONPs and PEE could effectively inhibit the growth of *S. aureus* and *E. coli.* The film of PLA/ZnONPs/PEE_0.5_ was selected for packaging test. The freshness-keeping test of cherry tomatoes showed that the shelf-life of cherry tomatoes was significantly prolonged by PLA/ZnONPs/PEE film. In conclusion, the addition of PEE and ZnONPs in PLA successfully endowed the PLA films with antioxidant and antibacterial activities, which made the PLA-based films have greater potential in food packaging applications. In future research, the cost of the bio-based composite film needs to be reduced for better application.

## Data Availability

Data are contained within the article.

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
