# Peer review of "Development of Pomegranate Peel Extract and Nano ZnO Co-Reinforced Polylactic Acid Film for Active Food Packaging"

_membranes, 2022, doi:10.3390/membranes12111108_

Round 1

Reviewer 1 Report

The manuscript is focused on the development of pomegranate peel extract and nano ZnO2 co-reinforced polylactic acid film for active food packaging. The idea is fairly novel and the experimental design was generally well-done and well organized. However, there are some serious shortcomings in the manuscript and I can not suggest the paper for publication in a high-standard journal like Membranes.

Here are some of the main flaws and remarks:

General comments: 

·      There are lots of grammatical mistakes in whole parts of the manuscript that make the text hard to follow and understand. So, it is strongly suggested that the manuscript must be revised by a native-speaker English editor.

·      The introduction is not fluent and is not focused. It should be completely revised.

·      The results were weakly discussed and they were not deeply analyzed.

Introduction:

Line 26: A space must be inserted between words and “(“ or “)”. Please correct here and elsewhere in the text.

Line 29: what is polylactic acid assistance?!

Line 33: replace “biobases’ with “biobased”.

Line 36: replace “universally recognized safe” with “generally recognized as safe”

Line 38: what is “philicity”?

Lines 39-42: Revise the following sentences: “lot of work has been done to improve its performance to make PLA more competitive with other commercial plastics. Such as the addition of plasticizers, polymer blending, conjugation agent, copolymerization, and nanotechnology”.

Line 45: “ZnONPs” and other abbreviations in the text must be defined at the first mention. Correct here and elsewhere in the text.

Line 46: revise “wide effective antibacterial period”

Line 51-52: correct and revise “the main antimicrobial active”

Line 53: what is the relationship between biological properties and antioxidant effects?

Line 58: replace “they’ with “it”

Line 62: add “properties” after “other”

Scheme 1: what is the function of this scheme and where is it cited in the text?

The objectives are too wordy and there is no need to mention all of the applied tests.

Line 72: specify the tested food instead of using “foods”.

Materials and methods

Line 80: correct “and were”

Line 81-82: concerning scientific names like the name of bacteria, they must be italic. Correct here and elsewhere in the text.

Line 83: delete this line

Lines 91-93: For explaining methods/processes and results in scientific writing, the passive voice must be used. Correct here and elsewhere in the text

Line 99: “was collected” is not fit here. Revise.

Line 106: replace “measured” with “studied”

Line 124: it is recommended to use “elongation at break” instead of “fracture growth rate “

Line 135: concerning the “water solution”, it is “moisture content”. Please correct.

Line 144: replace “water solution” with “water solubility”

Line 155: it is probably “the following equation”.

Lines 179-180: the parameters A1, A2 and A3 were explained earlier in the text. So, there is no need to define them twice.

Lines 195-196: delete Gram-positive and Gram-negative before the name of bacteria.

Line 203-204: What do you mean “both negative and positive controls”.

Line 205: delete “with separately”

Section 2.6: it must be explained how the apparent picture, weight loss, pH changes, and stiffness of the cherry tomatoes were monitored.

Add a section about the applied statistical analysis.

Result and discussion

Section 3.1.1: The results have not been deeply discussed, and they must also be compared with the data obtained in similar studies.

Figure 1: The caption is incomplete and the cross-section and surface pictures must be separated with some elements.

Table 1. use “mm” instead of “nm”; it seems that the statistical analysis was not done correctly.

Line 310: use “the” instead of “The”

Lines 331-332: revise the “and films with minimal WVP reduce water transport by preserving food, thus increasing the shelf life of food”

Line 33-334: figure not “table”

Lines 339-343: what is tightness? tightness increased the WVP or decreased it?!

Line 345:  revise “the activity of based on PLA”.

Figure 4: correct “blak” in the figure a.

Lines 372-377: There is no discussion about the effect of ZnOPs and PEE and why the addition of PEE expand the antibacterial effect of the films.

Line 381: film?

Line 393: It was mentioned a severe bacterial infection was observed! How do you detect the presence of bacteria?

Line 395: correct “indicatee”

Lines 395-396: it was mentioned that “these results indicate that PLA/ZnONPs and PLA/ZnONPs/PEE0.5 films 395 had a physical barrier effect on microorganisms”. It is more probably related to the antibacterial properties of the developed composites, as indicated in antibacterial tests.

Section 3.4.1.: The results have not been deeply discussed, and they must also be compared with the data obtained in similar studies. Moreover, it is better to report the weight loss in percent.

Good luck.

Reviewer 2 Report

In this study, the authors developed active PLA-based films containing ZnNP and pomegranate peel extract. In general, I believe that it is a very interesting work and, moreover, it is very complete in terms of characterization and evaluation of the results. This reviewer recommends its publication after some revisions, which are summarized below:

L37: the authors stated “... However, pure PLA materials also have some disavantages, such as poor philicity, poor mechanical properties...”. What do you men “philicity?”. Furthermore, the authors speak here that pure PLA has poor mechanical properties, but previously compared them with the properties of polymers from fossil sources. I believe the properties of PLA are good and should be highlighted. Rewrite this sentence for better understanding by readers.

L42: The authors stated “...Nanometalals are considered to be one of the most direct and effective strategies to improve antimicrobial and UV properties...” What about improving mechanical or permeation properties??

L53-62: The authors mentioned a little about active films, but the text is still a little poor in information. It would be good to cite examples of other components that could generate active films, such as essential oils and other antioxidant extracts. I suggest that the authors expand this part of the introduction by citing other examples of biopolymers with PEE and other active compounds. Some exemples of articles that should be cited in the text to broaden the discussion: https://doi.org/10.3390/ma14123305 ; https://doi.org/10.1016/j.foodhyd.2022.107746 ; https://doi.org/10.1016/j.foodhyd.2022.107620 ; https://doi.org/10.1007/s11694-022-01337-x ; https://doi.org/10.1016/j.lwt.2022.113124

L91: “... pour the solution evenly on the glass plate...”. What is the mass of film-forming solution per unit area of the plate? Was a glass Petri dish used? what is the diameter?? Add more info.

L108: “... The crystallinity was analyzed using an X-ray dif- 108 fraction (Bruker D8 propulsion A25, Germany)(Chae, Yang et al. 2002)...”. Please add more information on the analysis conditions to make it easier for the reader to reproduce the test. What is the measured angle (2Theta)? scan rate, etc.?

P248: “... The crystallographic properties of the pure PLA, PLA/ZnONPs, and 248 PLA/ZnONPs/PEE2 thin films were investigated by XRD analysis. As shown in the Fig. 249 2b, the pure PLA had a positive-squared crystal structure (α-form) and consists of 250 103-helical chains...”. Authors should identify which samples are in the indicated graph, as it is unclear.

In Figure 3, some data do not have the letters representing the statistical analysis. Put them and discuss the results better.

Reviewer 3 Report

The Manuscript entitled "Development of pomegranate peel extract and nano ZnO co-reinforced polylactic acid film for active food packaging" describes preparation, characterization and application of active films based on polylactic acid. Commercial ZnO nanoparticles and pomegranate peel extract were added as active matters. The films were characterized in terms of morphology, thermal, optical and mechanical properties, water vapor permeability, contact angle, as well as antioxidant and antibacterial activity. Finally, the potential of the synthesized films to preserve food during shelf-life was evaluated on cherry tomatoes. The topic of the manuscript is actual and relevant. Generally, manuscript is well structured and objectives are clearly stated. However, substantial improvements are needed. The main complaint is lack of discussion. Results are mainly just reported without explanation, comparison with literature data, highlighting the best composition etc. This especially applies for Evaluation of cherry tomatoes preservation. Also, numerical values should be mentioned in the text throughout the manuscript, not only in tables and figures. For example, mechanical properties are presented only in the figure. Although the trend of TS is analyzed in regard with ZnONPs and extract content, no values are reported in the text. Moreover, it is unclear if the such properties are suitable for food package application. Comparison with other similar films (reported in the literature and/or commercial materials) is needed.

Minor comments:

1. Abstract should be rewritten to include numerical values of the most important results

2.  Information about cherry tomatoes is missing in the Materials part. Also, the process of coating is not clear.  

3. bacterial names should be written in italic

4. figure 2b: legend is missing

5. line 288: curcumin?

6. Fig 3 is too small. I suggest removal of c, d, f, and g into separate fig

 7. line 393-4: The authors state that severe bacterial infections were observed in the unpacked and and PLA groups. The samples presented on the picture seem like infected by molds rather than bacteria

8. You have prepared 6 different formulations, but for XRD, thermal analysis and for coating cherry tomatoes only 3 were chosen. How did you select the film formulation? Please include the explanation for all test

9. Conclusion should be rewritten. Try to emphasize the specific contribution of your study and to highlight the most important findings. 

Round 2

Reviewer 1 Report

Dear authors,

The manuscript has been revised substantially. However, there are only a few more remarks that should be considered:

Section 3.1.1. Revise the "the surface morphology and cross-section changes of the composite film were thin and small, and the surface PEE particles were uniformly dispersed in the PLA/ZnONPs matrix". 

Correct "figure 45".

Figure 5.a. Correct "Blak".

Section 3.4. Revise "the results (Figure S1) showed that the unpackaged and PLA-packed groups grew more microorganisms than the other two groups". 

It is suggested again that the manuscript should be revised by a native-speaker English editor.

Reviewer 3 Report

The authors have corrected the manuscript in line with my suggestions. However, some issues remained unanswered. Although the authors addressed most of my questions in the author response file, some of those informations should be included in the manuscript. I hope that my suggestions  (listed below) now will be more clear and that will help the authors to improve the manuscript:

1. description of coating/wrapping tomatoes is missing

2. more discussion regarding different formulations is needed throughout the manuscript. Try to focus on materials' desired properties (for each analyzed characteristic) and discuss which formulation meet them best. Is the same formulation always the best choice? Is some compromise needed to fulfill all required properties? Does it depend on future application? These questions are only few examples on how to improve the discussion part of the manuscript.

3. Mechanical properties are presented only in the figure. Although the trend of TS is analyzed in regard with ZnONPs and extract content, no values are reported in the text. Moreover, it is unclear if such properties are suitable for food package application. For example, the authors state "EB gradually increased with increasing PEE content", but it is indistinct if this was good and desirable or not. Comparison with other similar films (reported in the literature and/or commercial materials) would also be helpful.

3. How did you chose the formulation for testing on tomato? The explanation should be included in the manuscript. Microbiological assay of tomatoes should also be described.

4. Conclusion is still weak. Try to answer the following questions: According to your results, what was the best formulation? Could it be selected at all?What is the specific contribution of your study? Is your material superior in comparison to all those already existing and previously tested? In which way? What are the limitations that still need to be addressed before such material could be applied in the industry? 

Minor comments:

1. You use three different terms for the same characteristic throughout the manuscript (fracture elongation, elongation at break, fracture growth rate). Please uniform.

2. line 510: you didn't test the biodegradability of the prepared films, so you cannot emphasize this feature in the conclusion.

Round 3

Reviewer 3 Report

l 211 - 213: the sentence is not clear, please check the meaning

l 418: "relatively excellent" is inappropriate term, please specify the values
